# A Novel Dynamic Graph Architecture for Staging Parkinson's Disease Progression Using Cerebrospinal Fluids Longitudinal Profiles

## Abstract

Dynamic graph learning methods typically capture local structural information and short-range temporal dependencies at each time step. In this work, we introduce a dynamic graph learning architecture that generates time-step embeddings capturing both local structural context and progression-trajectory patterns for each node across an entire longitudinal sequence. Unlike existing approaches, our framework clusters fused embeddings that integrate (i) the global temporal trajectory of each node and (ii) its local spatial context at every graph snapshot to discover meaningful temporal patterns in longitudinal datasets. We evaluate the proposed model in the context of Parkinson's disease (PD) progression using six years of longitudinal cerebrospinal fluid (CSF) profiles from 24 patients. Visit-based graphs were constructed by representing patients as nodes enriched with peptide-abundance features, and by connecting patients with similar features profiles. A Graph Convolutional Network (GCN) captures visit-specific spatial relationships, while a sequential model learns global temporal representations. A fusion module integrates both sources of information to produce enriched node embeddings that reflect inter- and intra-patient molecular dynamics. Clustering the learned embeddings reveals four distinct PD progression stages, supported by strong validity indices (Davies–Bouldin: 0.169; Calinski–Harabasz: 1264.24). Significant differences in motor severity (UPDRS_2 and UPDRS_3; $p < 0.05$) were observed across clusters, whereas non-motor scores showed a more diffuse pattern ($p = 0.11$). Compared with PCA, autoencoders, GCN, T-GCN, and GC-LSTM, the proposed architecture yields more clinically discriminative representations of disease severity. These findings demonstrate the potential of the proposed dynamic graph learning for data-driven disease staging and offer a generalizable framework for uncovering latent temporal patterns in longitudinal datasets.

## 1 Introduction

Graph Neural Networks (GNNs) have emerged as powerful tools for learning from graph-structured data by capturing nodes' local and global dependencies. Unlike traditional deep learning models, GNNs are not restricted to grid-like or strictly sequential input limits. This makes them well-suited for tasks involving complex and time-variant relationships among nodes features - dependencies that are often overlooked by conventional deep learning models such as Long Short-Term Memory (LSTM) networks and Convolutional Neural Networks (CNNs) de Jong et al. (2019); Çağatay Berke Erdaş et al. (2021); Gan et al. (2021); Li et al. (2021). For example, Dynamic Graph Neural Networks (DGNNs) have been developed to model every node's spatial and temporal variants in a sequence of graphs Wu et al. (2024); Zheng et al. (2024).

However, these generated embeddings often capture either local node relationships, such as embeddings learned by a Graph Convolutional Network (GCN), and/or short-term historical temporal patterns, like hidden state representations learned commonly by models like Temporal GCNs (T-GCN) and Graph Convolution Embedded LSTM networks (GC-LSTM). Although these approaches have shown effectiveness in supervised tasks such as node classification, feature prediction, and

edge inference, they have been underexplored in unsupervised representation learning tasks, and fall short particularly when modeling long-range spatio-temporal dependencies is required Zheng et al. (2024). While existing graph learning models are capable of capturing feature relationships within individual nodes, across nodes, and over time, a key limitation remains: their inability to include generalized temporal dynamics alongside local structural patterns to create comprehensive embeddings for each node at every graph snapshot.

Bridging this gap is critical for advancing unsupervised representation learning in longitudinal data, particularly for uncovering temporal progression patterns in abnormally evolving processes or conditions. Motivated by the GCN's ability to capture local spatial patterns within a single graph snapshot and the strength of sequential models in capturing temporal dependencies, this study proposes a dynamic graph learning architecture for modeling longitudinal records. Specifically, the model is designed to identify distinct temporal disease stages in the context of Parkinson's Disease (PD) progression case study. PD is the second most common progressive neurodegenerative disorder, with an estimated 11.77 million people worldwide in 2021 living with PD, and this number is projected to rise to 25.2 million by 2050 Luo et al. (2025); Su et al. (2025). PD abnormally evolves by time and manifests by motor symptoms and non-motor symptoms (such as mood changes), with an average survival of approximately nine years following symptom onset Ryu et al. (2023); Kouli et al. (2018). The exact cause and clinical trajectory of PD remain unclear and vary significantly between individuals due to the disease's inherent heterogeneity Balestrino & Schapira (2020); Abu Zohair et al. (2025).

Static graph models were increasingly applied to model data involving biomolecules, drugs, or patient-related features, such as clinical measurements or medication records Liu et al. (2025); Shang et al. (2019); Abu Zohair et al. (2025). While they have shown promise, they are inherently limited in capturing the temporal changes inherent in longitudinal data. In contrast, dynamic graph representation algorithms offer the ability to model evolving relationships over time and have been explored with some explainability efforts Wu et al. (2024); Zheng et al. (2024). However, to date, no prior work has effectively leveraged dynamic graph-based learning models to analyze longitudinal clinical features for understanding neural disorder diseases such as PD Zheng et al. (2024).

To uncover PD temporal disease stages, we constructed visit-based graphs representing yearly clinical visits of patients based on their cerebrospinal fluid (CSF) peptide profiles. The goal of the proposed architecture is to generate a meaningful representation for each clinical visit, allowing these representations to be clustered and revealing distinct disease stages. We hypothesize that this is feasible if the learned embeddings emphasize local structure while also capturing each patient's disease trajectory over time. To achieve this, the model employs a single-layer GCN to extract spatial representations for each patient at a given visit_month. These representations are then passed through a sequential model, such as a Gated Recurrent Unit (GRU), to learn generalized spatio-temporal patterns across time. The resulting temporal patterns are concatenated with month-specific spatial embeddings, and the combined representation is passed through fusion layers to produce node embeddings at each time step that capture both local structural dependencies and temporal progression.

The quality of the learned node representations by the proposed approach was evaluated by comparing the clustering performance of embeddings against several baseline methods, including standard graph representation learning models (such as GCN, T-GCN, and GC-LSTM) and conventional feature representation techniques, such as dimensionality reduction methods and dense autoencoders.

The main contribution of this study demonstrates that the proposed dynamic graph learning approach outperforms existing feature representation methods in modeling longitudinal clinical data. This means that incorporating the general temporal progression of the nodes emphasized by their local spatial embeddings enabled the discovery of meaningful stages of disease progression. To the best of our knowledge, this is the first graph-based model to generate such comprehensive spatio-temporal representations for unsupervised learning in this context. Furthermore, this work presents the first application of dynamic graph learning models to uncover meaningful stages of neural disorder disease, specifically in PD.

## 2 RELATED WORK

Existing graph learning models like GCN Kipf & Welling (2017) are designed for static graphs and cannot capture evolving structures over time. However, temporal extensions in dynamic graph models such as T-GCN Zhao et al. (2020) and GC-LSTM Chen et al. (2022) integrate temporal sequences with learned structural patterns but remain limited to capturing only short-term historical temporal dependencies when modeling current graph nodes. Moreover, the Temporal Graph Network (TGN) Rossi et al. (2020) generates node embeddings through temporal message passing with memory modules. However, in its common edge-based implementation, TGN does not utilise node features in the embedding process, nor does it compute node embeddings at each graph snapshot or capture long-range temporal dynamics in all its existing implementations. Models like EvolveGCN Pareja et al. (2019) and DyGCN Gu et al. (2024) update weights or node states over time, yet fail to jointly model local spatial features and generalized node trajectories across snapshots. To the best of the authors' knowledge, none of the existing approaches are designed for unsupervised representation learning in visit-based or longitudinal graphs, where each node evolves over time and must be embedded accordingly. The architecture is designed to explicitly capture both spatial structure and generalized temporal patterns per node, enabling more meaningful representations for downstream unsupervised tasks like disease stages discovery. This contribution remains largely unexplored in the current literature.

From a healthcare perspective, GNNs are not new and have been primarily employed for representation learning, graph pooling, and entities generations (like molecules represented as graph nodes or edges), for applications dominated by structural and semantic relationships understanding among biological entities like molecular interactions, protein functions, tissue-specific gene regulation, and predict disease- phenotype-disease associations Li et al. (2022). For example, subgraph neural networks (SubGNNs), portions of a larger graph, have been applied to model diseases as phenotype subgraphs derived from the Human Phenotype Ontology (HPO) knowledge base, for disease classification tasks Li et al. (2022). Additionally, researchers in Abu Zohair et al. (2025) structured each patient's CSF longitudinal records into a single graph and applied GCNs to model disease trajectories. However, an LSTM autoencoder was shown to outperform this approach. This approach and the aforementioned applications were based on static graph structures and lacked the employment of temporal modeling in a dataset of longitudinal nature. Also, in critical care and infection control, models like MSTD-GNN and STM-GNN have used temporal attention and memory to model irregular ICU records and hospital-acquired infection dynamics, respectively, but remain largely prediction-focused and rely on learning historical temporal patterns Geissbuhler et al. (2025); Liu et al. (2025). Researchers also integrates dynamic attention into CTDGs by fusing global medical ontologies with patient-specific knowledge graphs, adjusting by the graph edge weights in real time to improve disease diagnosis Chen et al. (2025).

Despite these advancements, notable gaps persist. Subgraph-based disease models in prior work do not account for dynamic symptoms evolution. Very few studies explore dynamic graph modeling, but not in the context of learning representation for unsupervised tasks. Discrete and continuous-time dynamic graph models were employed. However, their efficiency in handling incomplete longitudinal records has not yet been addressed. Finally, the potential of dynamic graph models in modeling longitudinal records for understanding disease dynamics progression in rare or neurodegenerative diseases has yet to be explored. This research contribution directly addresses these limitations by introducing a fused GCN-GRU architecture that models the influence of local structural patterns in every graph snapshot node while learning every node's temporal progression dynamics, enabling the unsupervised grouping of distinct disease stages across various progression trajectories. Finally, this work pioneers the first dynamic graph approach for modeling Parkinson's disease longitudinal records, offering insights into disease states that have never been captured in prior literature.

## 3 METHOD

### 3.1 DATASET

The dataset utilized in this study comprises longitudinal protein and peptide abundance profiles extracted from CSF samples collected from 24 patients, and was sourced from Kaggle Kirsch et al. (2023).It is acknowledged that the dataset is small and limits the generalisability of the findings,

however, it was readily accessible for demonstrating the proof of concept. In addition to CSF-derived peptide abundance data, another dataset that contains longitudinal clinical scores was utilized. From this dataset, we used UPDRS Parts 1, 2, and 3, which represent mood and cognitive symptoms, abilities for daily living, and motor complications before and after medication, respectively. Kirsch et al. (2023). Together, the CSF and UPDRS datasets support temporal analyses of Parkinson's disease progression, enabling the development of computational models to infer disease states, trace patient trajectories, and characterize patient-specific or disease-specific features within each identified stage.

## 3.2 METHODS AND PROCEDURES

### 3.2.1 CONVENTIONAL FEATURE REPRESENTATION METHODS

K-PCA and t-distributed stochastic neighbor (t-SNE), non-linear dimensionality reduction or latent representations of peptide types were selected as feature reduction techniques. This decision was made after preliminary analysis of data normality using the Shapiro–Wilk and Q-Q plots. In parallel, dense autoencoders were commonly outperformed for learning representations of patient records Sushil et al. (2018); Alkhayrat et al. (2020); Abu Zohair et al. (2025). In this project, a single-layer dense autoencoder was employed to learn the latent representation of patients' clinical visits. Following this, multiple graph representation learning models were explored. For example, a static GCN with a single convolution layer was applied to patient records structured in temporal clinical visit-based graphs, capturing only node structural patterns within a single graph. In addition, standard dynamic graph models, such as the T-GCN, extend the structural patterns (relationships among patient features within each clinical visit) learned by the GCN by incorporating connections to historical temporal patterns. In this work, a simplified implementation of T-GCN is presented, combining a single-layer GCN with a sequential model based on GRU. This design choice aims to avoid over-smoothing effects caused by aggregating higher-order adjacency neighbors, which can lead to homogenized node representations and the loss of node-specific local patterns. The following subsection presents the formal graph definitions and describes the novel architecture, which will be compared with the previously introduced approaches.

### 3.2.2 GRAPH STRUCTURE AND DEFINITIONS

To utilize graph learning representation models, the tabular representation of the dataset was transitioned into a graph-based structure. Inspired primarily by Kazemi et al. (2020), an undirected graph was constructed at each clinical visit in the longitudinal dataset (based on the `visit_month feature`), as illustrated in Figure 1. This graph is annotated by G=$(\mathcal{V}, \mathcal{E})$, where:

- $\mathcal{V}$ be the set of vertices (nodes), representing patients, indexed by the `patient_id` feature.
- Peptide types and their abundances for each patient are stored as node features and represented in a matrix $\mathbf{X} \in \mathbb{R}^{n \times d}$, where $n$ is the number of patients (nodes), and $d$ is the number of clinical features associated with each node.
- $\mathcal{E}$ be the set of edges added between nodes. An edge is added when the distance (measured by Euclidean distance) between two patients' features is below a specified threshold. This graph structure is represented by an adjacency matrix $\mathbf{A} \in \mathbb{R}^{n \times n}$, defined as:

$$A_{ij} = \begin{cases} 1, & \text{if the distance between patients } i \text{ and } j \text{ features} \leq \text{threshold} \\ 0, & \text{otherwise} \end{cases}$$

The threshold corresponds to the 5th percentile of the pairwise Euclidean distance distribution between node features. It was aimed to retain only the strongest and most meaningful connections, while reducing noise from weaker or less informative similarities. The final threshold was determined by averaging the 5th percentile of Euclidean distance distribution scores across all graphs.

The implemented architecture, illustrated in Figure 2, employs GCN to generate spatial node embeddings ($e_S$) at each graph snapshot, which are then fed into a GRU model to capture general spatial dependencies (within and across patients' features) and temporal dynamics (across visits), including current, historical, and future patterns ($e_t$). Finally, the learned spatio-temporal pattern for each

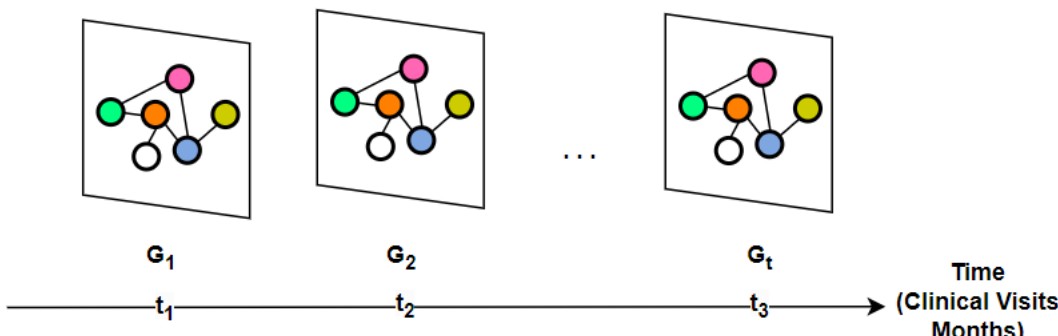

Figure 1: Temporal graph construction across clinical visits in the longitudinal dataset. Each node represents a patient's record at a specific clinical visit (based on the visit_month feature), with edges capturing the euclidean distance between the features of two patients' nodes.

node is concatenated with its spatial embedding at every clinical visit and passed through fusion layers to generate node representations ($e_{t+S}$), which are then used in an unsupervised clustering algorithm to identify disease states. It is worth noting that a multi-model implementation is also a valid approach, as the components chosen to capture structural-temporal embeddings versus structural embeddings may differ. In the current implementation, these components are integrated into a single model (GCN + GRU generate output, then the fusion of GCN and that generated output). However, this framework is flexible, and it would be possible to explore alternative architectures - one model for spatio-temporal representations and a different one for structural representations.

To define, the implemented models' architecture creates latent node embeddings for nodes at each visit month using a single layer of GCN encoder (GCNConv layer- graph convolution layer provided by PyTorch Geometric) from the standard implementation of static GCN Kipf & Welling (2017), Jiang et al. (2019). These embeddings are created by simply convolving over a node's direct neighbors, as defined by Equation 1.

$$H_t = \sigma(\hat{A}_t X_t W), \tag{1}$$

where: $H_t$ are the node embeddings of graph at clinical visit $t$, $\hat{A}_t$ is the normalized adjacency matrix at clinical visit $t$ , $W_t$ is the learnable weight matrix at clinical visit $t$, and $\sigma$ is the activation function (ReLU) that combines the graph structure and node features $X$ to compute the updated node embeddings.

After that, the spatial embeddings ($e^s$) created for each node at each time-step ('visit_month') were fed to a single GRU layer to create a single spatio-temporal embedding for the dynamics of each patient's features. It is worth noting that this implementation of T-GCN is slightly modified from its standard implementation by Zhao et al. (2020), with a reduced number of GCN layers used to compute the spatial nodes' embeddings. Additionally, the node embedding from each GRU hidden state for each node at each visit month (which accounts for both current and historical temporal patterns) was also extracted to assess its representativeness in discovering the disease state compared to the proposed one. By definitions, for each patient (node $n$), the spatial embeddings over time form this sequence $H_{n0}, H_{n1}, \ldots, H_{nt}$, where t denotes clinical visit time. The spatial embeddings for each node are then fed into a GRU and produce a sequence of hidden states $Z_{n0}, Z_{n1}, \ldots, Z_{nt}$ where $Z_{nt}$ is the GRU's hidden state for node n at clinical visit $t$. This is defined in Equation 2;

$$Z_{nt} = \text{GRU}(H_{n0}, H_{n1}, H_{n2}, \ldots, H_{nt}) \tag{2}$$

The spatial-general temporal embedding, annotated by $e^t$ in Figure 2, created for each patient (n) at last clinical visit (t) defined as $Z_n = Z_{1t}, Z_{2t}, Z_{3t}, \ldots, Z_{nt}$. This will be used to create the final node embedding, $e^{s+t}$ in Figure 2, for each node $n$ at each time step ($t$) by fusing it with the spatial nodes' embeddings created at each clinical visit month ($H_{nt}$) by Equation 3.

$$F_{nt} = \text{Fusion}([H_{nt} \parallel Z_{nt}]), \tag{3}$$

where: $\parallel$ denotes concatenation, and $F_{nt} \in \mathbb{R}^{N \times d}$ is the proposed model embedding for all nodes at every visit month $t$. `Fusion` corresponds to the fusion layers (illustrated in Figure 2), which

apply a fully connected layer to the concatenated embeddings $H_{nt}$ and $Z_{nt}$, projecting them into a fixed fusion dimension (`fusion_dim`). A ReLU activation is applied to the output to introduce non-linearity and help capture complex relationships. Then, it is finally passed through a normal-ization layer (LayerNorm) to stabilize training and improve convergence. The used Fusion layers were frequently employed when combining embeddings of features from multiple models or data modalities resources in the literature Liu et al. (2020), Li et al. (2017). The selection of these layers was done progressively, while assessing model training loss simultaneously, resulting with the final configuration chosen based on achieving the lowest model loss. The explained model components

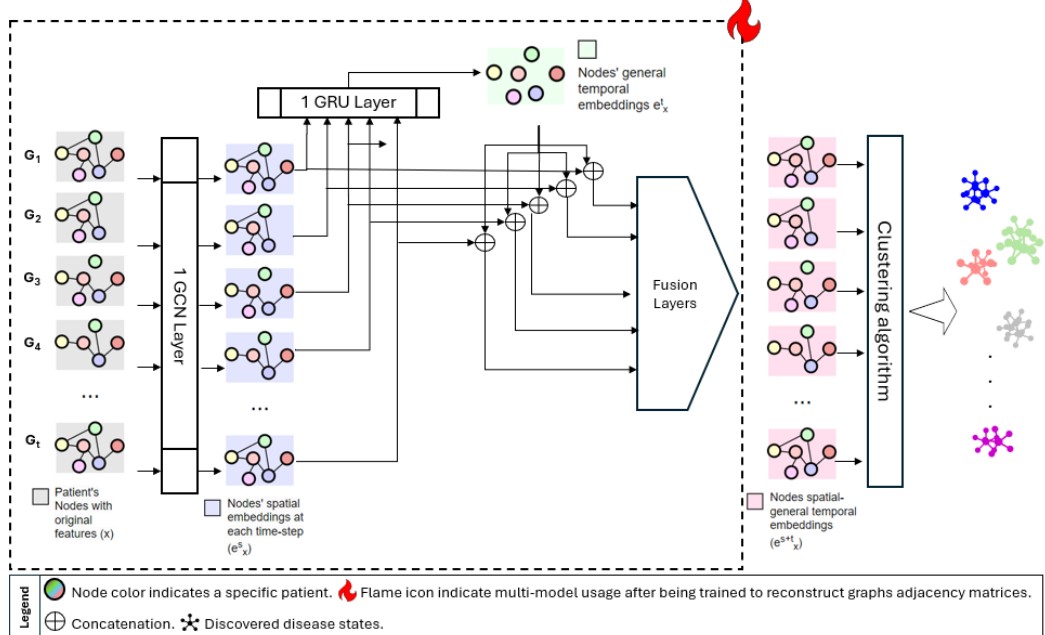

Figure 2: Overview of the Proposed Dynamic Graph Model Architecture. $G_{1-t}$ represents the graphs constructed at each unique visit_month in the longitudinal clinical visits' dataset. Each node color represents a unique patient who attended a clinical visit at the corresponding month. Missing nodes indicate time points where patients did not attend a clinical visit. The letter $x$ to denote original features, $e$ for features' embeddings, where superscript $S$ refers to the spatial embeddings and superscript $t$ denotes the temporal embeddings generated for each patient

were trained (surrounded by dotted rectangles with flame icon in Figure 2) to reconstruct the graph structure using a binary cross-entropy loss over the predicted adjacency matrices Kipf & Welling (2016). Once trained, node embeddings were computed by the model for each node at each visit graph; illustrated by the graphs with a pink background in Figure 2. These embeddings, as well as the T-GCN hidden state embeddings ($Z_{nt}$), which capture both current and historical temporal patterns in addition to structural information, were passed to the clustering phase. Clustering perfor-mance will determine the representativeness of the generated embeddings compared to conventional embedding learning approaches or direct feature clustering in discovering distinct groups of patterns that correspond to different but meaningful PD disease states.

### 3.2.3 EXPERIMENTAL DESIGN

The analysis started with visualizations and descriptive statistics to understand feature characteris-tics, including general temporal trends in peptide counts, UPDRS score progression patterns, and available patients clinical records per visit. Patients with richer longitudinal data were filtered and se-lected to better capture peptide trends over disease progression. Dataset pre-processing techniques, including zero imputation of empty peptide abundance, label encoding, and features' normalization using RobustScaler (known for better outliers handling Izonin et al. (2022)), ensured the data was normalized and suitable for both classical and deep learning models.

To further examine and validate the underlying structure and distribution of the data before selecting appropriate dimensionality reduction techniques and suitable statistical analysis methods, a series of statistical tests was conducted. Shapiro–Wilk and visualization of Q-Q plots were used to assess the data normality. Therefore, kernel principal component analysis (K-PCA) and t-distributed stochastic neighbor (t-SNE), non-linear dimensionality reduction or latent representations of peptide types, were selected as feature reduction techniques. It is worth noting that t-SNE is not a standard embedding method. However, we used t-SNE to assess cluster separability of discovered nonlinear patterns and as an exploratory baseline. Additionally, the normality test results in choosing the Kruskal–Wallis test to evaluate the significant differences in the distribution of UPDRS 1, 2, and 3 across the discovered disease states. Dunn's test was then applied to identify which cluster exactly differed significantly from each other in terms of UPDRS score. The formulated null hypothesis states that there is no significant differentiation among the discovered disease stages and UPDRS scores distribution, and a p-value threshold of 0.05 was set to reject it. Collectively, these statistical evaluations provided a robust foundation for selecting both classical feature reduction methods and assessing the significance of the discovered temporal pattern groups.

The proposed model architecture was implemented and was trained transductively on our limited dataset, meaning that all graphs were used to learn embeddings during training. Training was performed iteratively while performing a grid search to identify the best hyperparameters, including the latent dimensions for the GCN, GRU, and fusion layers, using the average adjacency reconstruction loss across visit graphs as the objective. After training, the model was switched to evaluation mode, and embeddings for each visit graph were extracted.

$KMeans++$ clustering algorithm was used, known to provide statistical assurance on the quality of the clustering Li et al. (2023). The quality of the clusters decides the optimal representation model performance, and is evaluated using 1) clustering metrics: Silhouette Score, Davies–Bouldin Index, and Calinski–Harabasz Score, and 2) the aforementioned significant test in the formulated hypothesis. The best number of clusters was selected based on the grid search $k$ value that results in the best Silhouette score.

## 4 RESULTS AND DISCUSSIONS

A total of 24 patients with up to 6 years of clinical visits and 968 unique peptide types were selected for the analysis. Initially, the original peptide values were directly clustered. Then, dimensionality reduction and embedding methods were employed to better capture the underlying structure of peptides progression patterns. Specifically, t-SNE, K-PCA, and dense autoencoders, with 2-dimensional embeddings that consistently achieved the highest clustering scores across evaluation metrics. In addition, graph- and sequence-based neural models were applied to capture more complex patterns within a single patient's peptides, across patients in a single visit, and the progression of peptide expression over time. These included GCN, T-GCN, GS-LSTM, and the new architecture. Across all graph representation learning models, the optimal hyperparameters were commonly found to be a GCN embedding dimension of 8 and a GRU hidden dimension of 4, which yielded low training loss and strong clustering performance.

According to Figure 3, both the proposed model architecture and GCN architectures achieved the highest clustering scores, but the proposed consistently converged to 4 clusters, indicating a stable temporal disease state representation. Instead, the GCN lack stability in clustering performance across various seeds. The seed was initially set with a fixed random value to ensure that all sources of randomness, especially those caused by model weight initialization, behave deterministically across processing. This guarantees that the graph model's training, evaluation, and inference are fully reproducible, which was critical for fair comparisons.

However, model stability was a critical factor in evaluating the reliability of learned representations for disease stage discovery. The variability in clustering results was assessed across 10 different seeds for both the GCN and evaluated architecture. As shown in Table 1, the variability analysis based on standard deviation indicates that, although the clustering performance remained relatively stable, the GCN model's estimated number of clusters ($K$) fluctuated between 3 and 9. This was accompanied by substantial variation in the corresponding Calinski-Harabasz (CH) score. In contrast, the proposed architecture demonstrated markedly greater stability—consistently producing 4 clusters and yielding relatively stable clustering scores across all seeds (see Table A1).

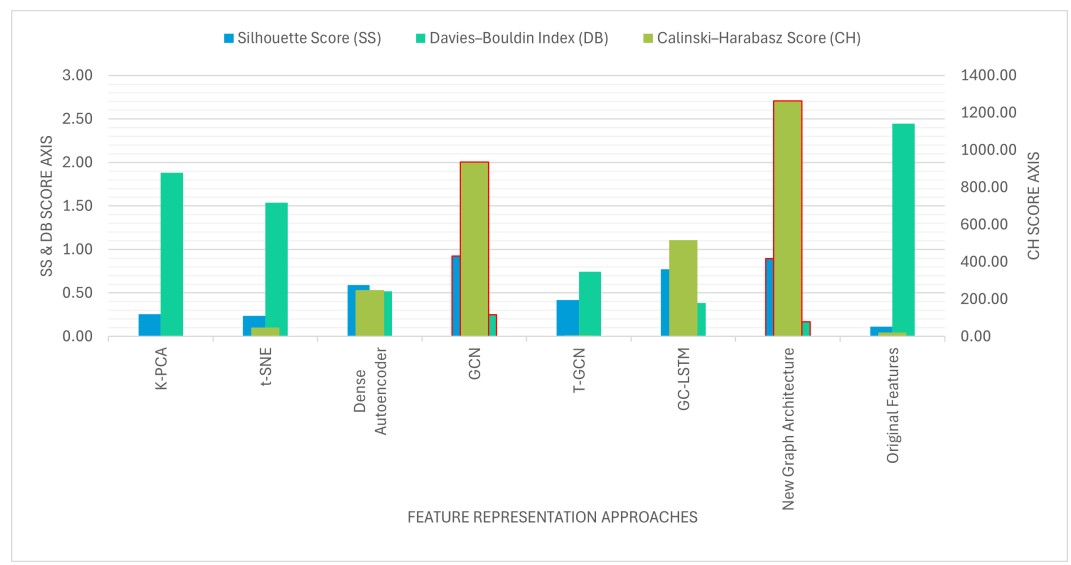

Figure 3: Clustering performance metrics results of peptides structural and/or temporal representations created by multiple feature representation Methods, including the new graph architecture.

Table 1: Standard deviation of clustering metrics across 10 different seeds for GCN and the new proposed model architecture. Lower values indicate higher stability.

| Metric/Clusters | GCN (SD) | Proposed Dynamic Graph Architecture (SD) |
|---|---|---|
| Silhouette Score (SS) | 0.02 | 0.04 |
| Davies-Bouldin (DB) | 0.08 | 0.14 |
| Calinski-Harabasz (CH) | 57,110.58 | 323.84 |
| Number of Clusters (K) | 1.79 | 0.46 |

To explain the disease stages from clinical perspectives, patients peptides dataset, including the discovered cluster, was merged with the UPDRS scores dataset - both originating from the same data source - using patient_id and visit_month as keys. Then, clusters formed by the new model architecture, which demonstrated strong internal validity as indicated by its Davies-Bouldin (0.169) and Calinski-Harabasz indexes (1264.24), were analyzed. To facilitate this and unlock the severity of disease stages based on these scores, a box plot of the distribution of UPDRS scores across the discovered clusters was generated (see Figure 4).

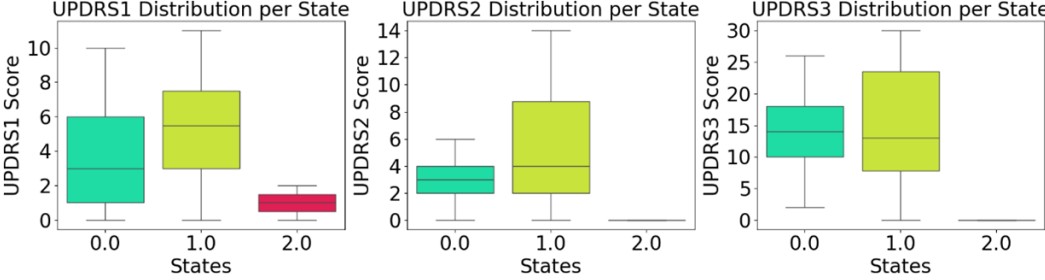

Figure 4: Box plot for the distribution of UPDRS 1, 2, & 3 scores across the discovered clusters.

It is important to note that one cluster appears missing in this figure. This was due to the exclusion of patients without corresponding UPDRS scores during the dataset merging process. As a result, (cluster 4) are not represented in the final UPDRS-based analysis. Kruskal-Wallis test revealed sta-

tistically significant differences (p ≤ 0.05) in both UPDRS Part 2 (motor experiences of daily living) and Part 3 (motor examination) scores across the plotted clusters. These differences are illustrated in Figure 4. A post-hoc Dunn's test was conducted to determine pairwise differences between clusters. This analysis, with  5, reveals that UPDRS Part 2 scores showed a significant difference between clusters 1 and 2 (p = 0.0306), with cluster 1 represent more sever UPDRS 2 symptoms compared to 2. For UPDRS Part 3, the differences were more pronounced, with cluster 2 exhibiting distinctly lower motor scores compared to clusters 0 and 1. In contrast, UPDRS Part 1 (non-motor experiences) was more evenly distributed across three clusters, but did not reach statistical significance (p = 0.11). This indicates that while the clustering effectively defines disease stages based on motor

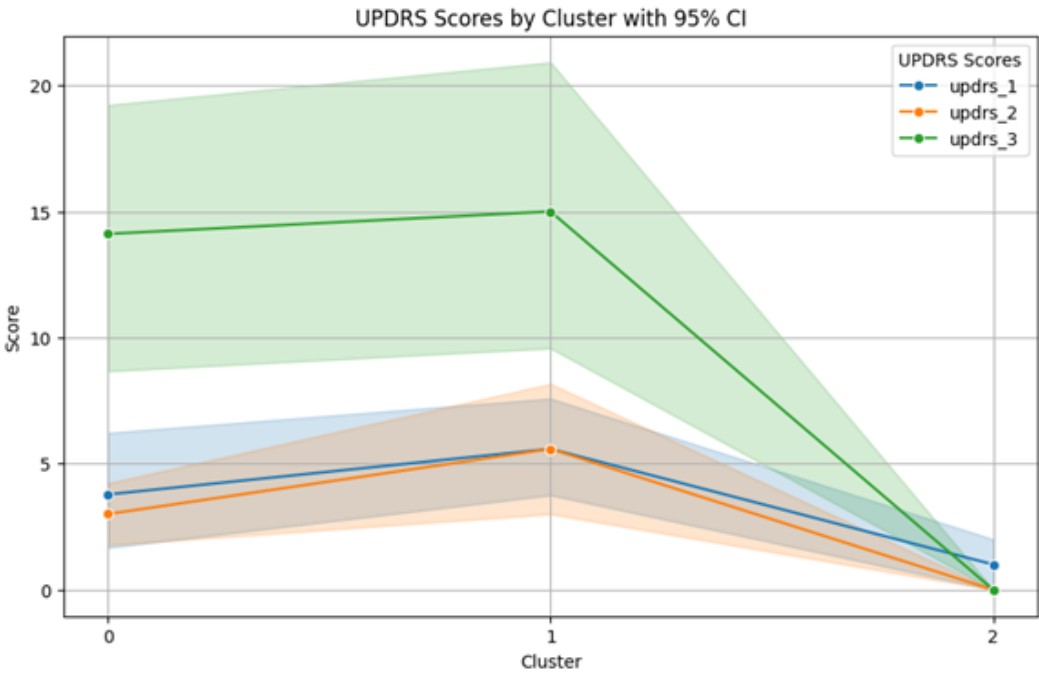

Figure 5: Mean UPDRS scores (Parts 1, 2, and 3) across discovered clusters, visualized with connected lines. Standard deviations are represented by shaded areas. This plot illustrates the variability in scores between the discovered clusters.

symptoms, it may be less sensitive to non-motor symptoms. These findings suggest that Cluster 1 reflects significant severe impairment in motor function and daily living activities for PD patients, while Cluster 2 groups those with mild symptoms of the disease. Despite the limited size of the dataset, the identified clusters capture clinically meaningful variation in motor symptom severity.

This work focuses on evaluating a generic model for grouping dynamic patterns in longitudinal records, rather than providing immediate clinical insights or patient-level interpretation. As a result, the clinical applicability of the findings is limited at this stage. Besides, the approach has not yet been validated on larger or external clinical cohorts, and we did not conduct a detailed error analysis to investigate misclustered cases and their underlying characteristics. These aspects represent important directions for future work and will yield more reliable group patterns, with statistical tests that provide more reliable and robust evidence. Future work will aim also further optimize the proposed architecture and by incorporating sequential models such as LSTM. In addition, comparisons between multiple linear perceptrons (MLPs) in place of GCN will be explored. This enables us to evaluate the actual contribution of the graph structure (adjacency matrix) to the learning process. Since MLPs do not use connectivity information, they serve as a strong baseline to evaluate whether leveraging structural relationships in GCN causes a performance gain or if it is largely due to node features. Additionally, alternative distance and similarity measures for edge construction, such as cosine similarity, will be explored. Furthermore, we plan to further assess the contribution of the fusion layers through additional ablation studies. Also, inductive model training with and external validation to more rigorously assess generalizability and avoid overfitting will be worth investigat-

ing. This proposed approach has demonstrated efficiency as a proof of concept, and future research will investigate its applicability to other neurodegenerative diseases as well as non-healthcare longitudinal datasets.

## ETHICS STATEMENT

This research utilized publicly available dataset and does not include human subjects, sensitive data, or applications with foreseeable potential for harm. To the best of the authors' knowledge, our research has no ethical concerns related to fairness, privacy, security, or legal compliance.

This manuscript was written using Overleaf, and the coding and analysis were conducted in Google Colab. During that time, various AI solutions were auto-suggested for text writing, code generation, and, not to mention, the facility for code debugging and error resolution. However, all generated content was critically reviewed, modified where necessary, and validated to ensure that it fulfilled its intended purpose and maintained scientific rigor. While using these tools to improve productivity and resolve technical issues, the author ensured that the main work, including critical reasoning, ideation, and interpretative decisions, remained fully under the author's control, thereby preserving the originality and integrity of this work.

## REPRODUCIBILITY STATEMENT

The dataset used in this study is publicly available and can be downloaded directly from https://www.kaggle.com/c/amp-parkinsons-disease-progression-prediction. All the code used for data preprocessing, analysis, and modeling is openly accessible at https://github.com/LubnaM/Graph_Multi-Model, ensuring full transparency and reproducibility of the results presented in this paper.

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

## A  APPENDIX

Table A1: Clustering scores across 10 different seeds for GCN and the new model's architecture. SS: Silhouette Score, DB: Davies-Bouldin, CH: Calinski-Harabasz, K: Number of clusters.

| | GCN | | | |
|---|---|---|---|---|
| **Seed** | **SS** | **DB** | **CH** | **K** |
| 0 | 0.9508 | 0.2548 | 2,458.85 | 7 |
| 12 | 0.9640 | 0.2675 | 8,622.10 | 6 |
| 22 | 0.9708 | 0.1234 | 13,888.18 | 5 |
| 32 | 0.9843 | 0.0278 | 189,078.22 | 4 |
| 42 | 0.9682 | 0.2481 | 6,095.28 | 6 |
| 52 | 0.9491 | 0.2777 | 4,819.29 | 8 |
| 62 | 0.9576 | 0.3202 | 1,422.66 | 4 |
| 72 | 0.9282 | 0.2829 | 915.69 | 5 |
| 82 | 0.9877 | 0.0790 | 4,690.02 | 3 |
| 92 | 0.9264 | 0.3313 | 1,716.94 | 9 |

| | New Graph Architecture | | | |
|---|---|---|---|---|
| **Seed** | **SS** | **DB** | **CH** | **K** |
| 0 | 0.8491 | 0.2606 | 467.83 | 3 |
| 12 | 0.8494 | 0.3491 | 695.68 | 4 |
| 22 | 0.7854 | 0.6502 | 252.38 | 3 |
| 32 | 0.8465 | 0.2366 | 322.02 | 4 |
| 42 | 0.8958 | 0.1694 | 1,264.24 | 4 |
| 52 | 0.9056 | 0.3463 | 645.35 | 4 |
| 62 | 0.7975 | 0.4103 | 406.15 | 4 |
| 72 | 0.9040 | 0.3430 | 843.84 | 4 |
| 82 | 0.8635 | 0.4895 | 307.84 | 4 |
| 92 | 0.7684 | 0.6719 | 205.83 | 4 |

