# OpenReview forum: "Unsupervised Dynamic Graph Multi-Model Representation Learning for Temporal Patterns Discovery: Uncovering Parkinson’s Disease Stages Using Cerebrospinal Fluid Longitudinal Profiles"
_ICLR.cc/2026/Conference — Submitted to ICLR 2026_

### Official Review · Reviewer_ZZtF · 2025-10-24

**Soundness:** 1
**Presentation:** 1
**Contribution:** 1
**Rating:** 2
**Confidence:** 5

**Summary:**

The paper presents a novel unsupervised dynamic graph multi-model framework for discovering temporal disease patterns from longitudinal biomedical data, with application to Parkinson’s disease (PD). The authors integrate a Graph Convolutional Network (GCN) for learning spatial dependencies among patients at each clinical visit with a Gated Recurrent Unit (GRU) for modeling temporal progression across visits. The model fuses spatial and temporal embeddings to produce comprehensive node representations, which are then clustered to reveal disease stages. Evaluation on a longitudinal cerebrospinal fluid (CSF) dataset of 24 PD patients demonstrates that the proposed model identifies four distinct disease stages, with significant correspondence to clinical UPDRS motor scores.

**Strengths:**

1. The combination of GCN (for intra-timepoint spatial modeling) and GRU (for temporal modeling) is well-motivated and implemented in a consistent framework.

2. The linkage between discovered clusters and clinically meaningful UPDRS scores provides valuable interpretability and supports the validity of the results.

**Weaknesses:**

- The manuscript is lengthy and sometimes reads as a descriptive technical report rather than a concise scientific contribution; key design motivations and insights are not emphasized. For instance, the abstract focuses too much on procedural details rather than emphasizing the core contributions, **resulting in an overly lengthy summary that obscures the main point.**
- Only 24 patients were included in the final analysis, which significantly limits the generalizability and robustness of the conclusions.
- The proposed architecture largely combines established modules (GCN and GRU) with standard fusion operations; no novel learning mechanism or theoretical contribution is introduced.
- The impact of each model component (GCN, GRU, fusion layers) is not quantitatively assessed. **The ablation analysis is necessary.**
- The paper does not provide a clear theoretical justification or complexity analysis explaining why the fusion of spatial and temporal embeddings improves representation quality.
- While ICLR permits supplementary material for code submission, the authors provided a GitHub link (seems not fully anonymized) pointing to their repository, yet no implementation code could be found there.

**Questions:**

Please refer to the above **Weaknesses**.

---

> ### Author Response · Authors · 2025-11-21
>
> Comments on the 1st and 2nd weaknesses:
> We acknowledge that the dataset of 24 patients is small and limits the generalizability of the findings. This dataset was selected because it was readily accessible demonstrating the proof of concept. As noted in the final paragraph of the discussion, the limited sample size is a recognized limitation, and future work will focus on evaluating the approach on larger and more diverse PD patient populations. The primary aim of this work is to assess a generic model capable of grouping dynamic patterns in longitudinal records, rather than providing an immediate clinical application or patient-level interpretation. For this reason, we submitted the paper in the “Learning on Graphs and Other Geometries & Topologies” track, rather than a healthcare-specific venue. But your point is valuable, and will be considered, particularly when extending the approach to larger or external clinical cohorts.
>
> Regarding the 3rd weaknesses:
> We understand the reviewer’s perspective regarding the novelty of our architecture.  To clarify, to group longitudinal patterns, we need to create a representation for each node at every visit-based graph. Each node’s embedding should capture not only its current local state but also the full progression trajectory it belongs to. If we were to extract hidden states from existing dynamic graph models, such as GC-LSTM, the temporal patterns would only reflect the node’s local and historical progression - not its future trajectory or the overall progression pattern. To our knowledge, no existing dynamic graph model captures such comprehensive spatio-temporal patterns in this way, which signifies the novelty of our approach.
>
> For your 4th point:
> This is an important comment; it would be interesting to evaluate the clustering performance not only for GCN embeddings and the full proposed architecture, but also potentially without the inclusion of the fusion layers. The future work has been amended to include this. See line [493-494].
>
> For the 5th point:
> Thank you for the comment. As described in lines [288–293], fusion layers were chosen to combine embeddings from multiple modalities, following established practices in the literature (Liu et al., 2020; Li et al., 2017). The selection of these layers was guided by monitoring model training loss, ensuring that the final configuration improves representation quality by effectively integrating complementary spatial and temporal information. While a full ablation and formal complexity analysis will be conducted in future work (as mentioned in our response to Q4), our current experiments already demonstrate the practical benefit of this fusion.
>
> Last, we have added the implementation code to the GitHub repository. The profile is anonymized, with a unique nickname, ensuring that no information or content could reveal author actual identity or affiliation.

---

> > ### Comment · Reviewer_ZZtF · 2025-11-26
> > **Thanks for the response.**
> >
> > Thank you for the response. I’m happy to know that you considered my comments valuable.

---

> > > ### Author Response · Authors · 2025-11-26
> > >
> > > We hope this clarification helps address any misunderstanding that may have contributed to the lower rating. If any additional revisions are required, please let us know. We still have time until December 3rd, and we would appreciate your feedback. Kindly review the attached revised PDF.

---

### Official Review · Reviewer_dDiY · 2025-10-27

**Soundness:** 3
**Presentation:** 2
**Contribution:** 3
**Rating:** 4
**Confidence:** 3

**Summary:**

This paper presents an unsupervised dynamic graph learning framework designed to discover temporal disease stages from longitudinal biomedical data. The authors propose a multi-model architecture that integrates a single-layer GCN, to capture spatial inter-patient similarity at each time step with a GRU to capture temporal dependencies across visits. The fused embeddings are clustered to identify Parkinson’s Disease stages using longitudinal cerebrospinal fluid peptide profiles. The method is evaluated against standard baselines using clustering validity metrics and non-parametric statistical tests. Results show that the model identifies four interpretable PD stages that correlate with UPDRS motor scores.

**Strengths:**

1. Using dynamic graph learning for unsupervised disease stage discovery is original, particularly in modeling temporal patient trajectories via age-based graphs.

2. The combination of GCN and GRU with a fusion module is conceptually coherent and technically straightforward. The reasoning behind using shallow layers to avoid over-smoothing is well-motivated.

3. The use of clustering validity indices and statistical significance testing (Kruskal–Wallis, Dunn) to validate discovered disease stages adds credibility to the biological interpretation.

**Weaknesses:**

1. The “multi-model” framework is effectively a shallow GCN + GRU fusion, a design already seen in many T-GCN and GC-LSTM variants. The contribution lies more in the application and evaluation context than in model innovation.
2. Only 24 patients are used after filtering, which raises concerns about overfitting and the reliability of clustering outcomes. The reported high Calinski–Harabasz and low Davies–Bouldin scores may not generalize.
3. The significance tests (e.g., Kruskal–Wallis, Dunn) are performed on very small sample sizes (some clusters appear to have few patients). The power of these tests and their reliability for clinical interpretation are questionable.
4. Presentation issues and overclaiming.
(a) The text occasionally conflates Parkinson’s and Huntington’s disease (see Section 2–3 confusion line 210–213), which undermines clarity. (b) Figures are referenced but not well-integrated in the discussion (e.g., Figure 2 architecture schematic is described verbosely but visually contributes little). (c) Claims of “first dynamic graph model for neural disorder diseases” seem overstated, given existing work in dynamic GNNs for biomedical longitudinal analysis.

**Questions:**

See weakness

---

> ### Author Response · Authors · 2025-11-21
>
> Comments on weaknesses 1:
> We understand how this might be confusing for first read, T-GCN and GC-LSTM typically process temporal dependencies over a fixed, often small, number of past graph snapshots, focusing on immediate historical patterns rather than the full longitudinal trajectory of each node. In contrast, our proposed architecture is designed to generate embeddings that capture both local patterns and the broader spatio-temporal progression across all visits. Empirically, we observed that embeddings produced by T-GCN or GC-LSTM showed lower clustering stability and less consistent capture of long-term disease progression compared to our fused GCN–GRU framework. If we were to extract hidden states from existing dynamic graph models, such as GC-LSTM, the temporal patterns would only reflect the node’s local and historical progression - not its future trajectory or the overall progression pattern. To our knowledge, no existing dynamic graph model architecture captures such comprehensive spatio-temporal patterns in this way, which highlights the novelty of our approach.
> Comments on weaknesses 2 and 3:
> We fully acknowledge the points raised by the reviewer. We acknowledge that the dataset of 24 patients is small and limits the generalisability of the findings. This dataset was selected because it was readily accessible for demonstrating the proof of concept. As noted in the final paragraph of the discussion, in lines 161-162 and 472-473, the limited sample size is a recognised limitation, and future work will focus on evaluating the approach on larger and more diverse PD patient populations. The primary aim of this work is to assess a generic model capable of grouping dynamic patterns in longitudinal records, rather than providing an immediate clinical application or patient-level interpretation. For this reason, we placed the paper in the “Learning on Graphs and Other Geometries & Topologies” track, rather than a healthcare-specific venue. But your point is valuable, and will be considered, particularly when extending the approach to larger or external clinical cohorts, it would be insightful to perform a thorough error analysis and investigate misclustered cases and their underlying characteristics. We included this in to the future work section in lines 474-478. Thank you for the helpful remarks.
>
> Regarding the presentation and overclaiming:
> (a) We appreciate the comment, and the text has been amended accordingly. (b) We thank the reviewer for pointing this out. There is an explanation intended for Figure 2, but the text mistakenly referred to other figures; the references in lines 269, 307and 310 have now been corrected to properly point to Figure 2. To the best of our knowledge, no prior work has applied dynamic graph models specifically to neurodegenerative diseases (NDDs) for the purpose of grouping longitudinal patterns. While dynamic GNNs have been explored in biomedical longitudinal data, these studies focus on other tasks but not for unsupervised learning purposes, and none explicitly address the combination of node-level spatio-temporal embeddings and longitudinal patient trajectory grouping in NDDs.

---

### Official Review · Reviewer_9Jj8 · 2025-10-30

**Soundness:** 1
**Presentation:** 1
**Contribution:** 2
**Rating:** 0
**Confidence:** 4

**Summary:**

This paper proposes an unsupervised graph-based learning framework for longitudinal disease data analysis. Each node represents a single patient, and edges encode pairwise similarity between nodes. A single graph is constructed based on the same aged patients, and a set of such graphs represents the temporal evolution of the cohort. The model integrates GCN and GRU to capture spatio-temporal dependencies across these age-based graphs, and a downstream clustering algorithm is applied to identify disease stages. The clustering quality is evaluated using three metrics, demonstrating that the proposed framework generally outperforms the baseline GCN.

**Strengths:**

1)	The motivation of the study, i.e., learning longitudinal patterns of neurodegenerative diseases, is clear and intuitive.
2)	The research includes statistical analyses of results.

**Weaknesses:**

1)	There are lack of technical novelty and significant lack of comparisons with existing works. The proposed method does not introduce a fundamentally new learning mechanism. The model is simply built upon a simple GCN with an additional GRU layer and a downstream clustering phase, offering incremental changes. Moreover, existing spatio-temporal graph models were not sufficiently examined; among the adopted baselines, only T-GCN is a spatio-temporal graph model. Additional recent baselines [1-4] are encouraged to be added. Given the growing number of graph studies, I believe that the authors could find more spatio-temporal graph learning methods with open source codes to strengthen the contribution of the proposed method.

[1] Cini et al., “Scalable Spatiotemporal Graph Neural Networks”, AAAI 2023

[2] Tang et al., “Predicting 30-Day All-Cause Hospital Readmission Using Multimodal Spatiotemporal Graph Neural Networks”, IEEE Journal of Biomedical and Health Informatics, 2023

[3] Cho et al., “Mixing Temporal Graphs with MLP for Longitudinal Brain Connectome Analysis”, MICCAI 2023

[4] Pareja et al., “EvolveGCN: Evolving Graph Convolutional Networks for Dynamic Graphs”, AAAI 2020

2)	Ablation studies of the model components and ranges of hyperparameter tuning (for both the proposed method and the baselines) are not provided. Including these details would improve the transparency and reproducibility of the experiments.
3)	Overall, the presentation of the paper has lots of room for improvement, including the writing, figures, and formulas. For example, the abstract contains too much details of experimental settings. The flow of the Introduction section is disorganized and mixed with background, motivation, and methodological contributions without a clear transition. In the method section, no loss function is given; instead, dataset description and model training configurations are presented, which would be more appropriately placed in the experiment section. Figure 2 is overly complex and difficult to interpret; it would benefit from a clearer depiction of the model architecture that highlights the key components and their interactions, rather than presenting every detail from all $G_t$. Furthermore, Fig. 3 does not provide exact numerical values of the performance metrics, making it difficult to compare the models quantitatively. Presenting these results in a table format would make the comparison clearer and more informative. The paper lacks an introduction of the evaluation metrics (e.g., UPDRS), which makes it difficult for readers outside the clinical domain to interpret the reported results.

**Questions:**

1)	Why is the method described as a multi-‘MODEL’ framework? The proposed method combines a GCN and a GRU, but GRU (Gated Recurrent Unit) itself is a unit/layer, rather than an independent model architecture. Moreover, although the loss function of the proposed method is not explicitly stated in the paper, it appears that the method only utilizes a single loss function, which makes it difficult to call it ‘multi-model’ framework.

2)	Why were edges constructed based on small nodal similarities rather than high similarities? I think the term ‘similarity’ might have been misused and should be replaced with ‘distance’, since Euclidean distance was employed as the measurement.

3)	Compared to recent spatio-temporal graph studies, what is the concrete technical advantage of the proposed method, and how do each model component meaningfully contribute to model performance or interpretability?

4)	In line 127, it is stated that T-GCN and GC-LSTM are limited to capturing only short-term temporal graph features. However, the rationale behind this claim is not clearly supported. It would be helpful if the authors could provide either a theoretical justification or quantitative comparison results demonstrating how the proposed method captures longer temporal dependencies compared to existing approaches.

---

> ### Author Response · Authors · 2025-11-21
>
> We understand your perspective in relevant to your 1st concern, and we fully agree with your claim, especially at its current setup. To clarify, the term “multi-model” in our original description stemmed from the conceptual idea of using separate components to capture the general structural-temporal embeddings for each node versus local structural embeddings for nodes at each visit-based graph. In the current implementation, these components are integrated into a single model (GCN + GRU generate output, then the fusion of GCN and that generated output). We agree that only one model is effectively used in this work. However, our framework is flexible, and in future work it would be possible to explore alternative architectures – for example, using one model for spatio-temporal representations and a different one (such as a spectral model) for structural representations - to further investigate embedding strategies. The manuscript is amended to exclude the definition of the multi-model for now. See amended paragraph starts from line 236.   We acknowledge that this terminology may have been misleading and will revise it for clarity. The framework remains flexible, and future extensions could incorporate truly separate models. Please also read our response to Q3.
> It is also worth noting that the term “multi-model” does not inherently require multiple loss functions.   In this work, we employ a single loss function because the model operates on one data modality and performs one specific prediction task - the reconstruction of the graph structure (adjacency matrix mainly). Contrastive or multimodal objectives are not required in this context. In multimodal or multi-task architectures, it is common to combine several loss functions to supervise different types of outputs. In addition, most graph representation learning methods that focus solely on reconstructing graph structure, like GAE and VGAE use only one loss function.
>
> Regarding 'Why were edges constructed based on small nodal similarities rather than high similarities? I think the term ‘similarity’ might have been misused and should be replaced with ‘distance’, since Euclidean distance was employed as the measurement.':
>
> Thank you for pointing this out. While more similar features naturally have smaller distances, using the term “similarity” could be misleading given that edges are defined based on distance. We agree that this terminology may cause confusion, and it has now been corrected throughout the paper to accurately reflect that distance, rather than similarity.
>
> And as a response to 'Compared to recent spatio-temporal graph studies, what is the concrete technical advantage of the proposed method, and how do each model component meaningfully contribute to model performance or interpretability?'
>
> This is partly explained in our answer to Q1. But to elaborate further, in order to group longitudinal patterns, we need to create a representation for each node in every visit-based graph. Each node’s embedding should capture not only its current local state but also the full progression trajectory it belongs to. If we were to extract hidden states from existing dynamic graph models, such as GC-LSTM, the temporal patterns would only reflect the node’s local and historical progression—not its future trajectory or the overall progression pattern. To our knowledge, no existing dynamic graph model captures such comprehensive spatio-temporal patterns in this way, which highlights the novelty of our approach.
>
> Finally, for 'In line 127, it is stated that T-GCN and GC-LSTM are limited to capturing only short-term temporal graph features. However, the rationale behind this claim is not clearly supported. It would be helpful if the authors could provide either a theoretical justification or quantitative comparison results demonstrating how the proposed method captures longer temporal dependencies compared to existing approaches.'
>
> Our response to 1st and 3rd questions covered some of the issues raised in Q4. Our claim that T-GCN and GC-LSTM are limited to capturing short-term temporal features is based on both design considerations and empirical observations. T-GCN and GC-LSTM typically process temporal dependencies over a fixed, often small, number of past graph snapshots, focusing on immediate historical patterns rather than the full longitudinal trajectory of each node. In contrast, our model is designed to generate embeddings that capture both local patterns and the broader spatio-temporal progression across all visits. Empirically, we observed that embeddings produced by T-GCN or GC-LSTM showed lower clustering stability and less consistent capture of long-term disease progression compared to our fused GCN–GRU framework, as illustrated in Figure 3 and reflected in silhouette score comparisons. These results indicate that our approach better preserves long-range dependencies in irregular, visit-based graphs

---

### Official Review · Reviewer_QbpR · 2025-11-01

**Soundness:** 2
**Presentation:** 2
**Contribution:** 2
**Rating:** 6
**Confidence:** 4

**Summary:**

This work proposes a new unsupervised dynamic graph representation learning framework for longitudinal biomedical data applied to a Parkinson's disease progression dataset. It combines per-visit graph neural network embeddings based on patient similarity edges, yielding integrated patient representations that capture both within-visit context and the entire across-visits temporal trajectory. It is claimed that this method outperforms methods like PCA and autoencoders, among others, while also claiming that the clusters discovered are meaningful.

**Strengths:**

1. It tackles the unsupervised representation learning problem for temporal graph-structured data in healthcare, an area that has been underexplored.
2. The architecture seems to be a clever/innovative way to put together previous well-established GNN methods for the specific challenge at hand.
3. The model naturally handles missing visits by simply omitting nodes for those missing visits (i.e. if a patient has no record at a given timepoint, they are not included in that snapshot graph). This approach to incomplete longitudinal data is practical and avoids complex imputation, while I also think it's a clever way to handle missing visits that I have not seen before.
4. The resulting clusters appear reasonable.
5. The method seems to achieve the highest clustering scores across many different baselines
6. The paper reports that the clusters identified by the so-called Multi-Model are more stable than the GCN baseline, which is important.

**Weaknesses:**

1. The experiments are run on a dataset with only 24 patients which is quite limited and thus raises concerns about robustness. The cluster findings might not generalise well for broader PD populations.
2. It's unclear if this model would scale to larger patient cohorts, as training multiple GCNs with so many nodes and a sequence model could be computationally heavy and not handle hundred of patients as seen in other datasets without significant modification or risks of overfitting.
3. Some aspects of the writing are confusing or inconsistent, which impacts clarity. For example, the authors describe constructing “age-based” graphs, when in fact each graph corresponds instead to a specific visit time (a better -and correct- term in my opinion would be visit-based graphs). The term “multi-model” is overused without a clear definition, and in my perspective only one model is actually being presented in this work, rather than many. Additionally, section 3.1 wrongly refers to "both datasets" even though only one dataset is used, suggesting a leftover from a previous draft. Finally, I believe "Huntington's disease" at the end of Related Work is a typo.
4. The inclusion of t-SNE as a baseline for the embedding/clustering comparison is questionable. t-SNE is a stochastic visualisation algorithm, not a fixed representation learning method, so its results can vary run to run and it's not typically used for clustering performance benchmarks.
5. The work lacks any evaluation on external datasets or discussion of generalisability. All results are on a single small PD cohort, and there's no evidence the learned representations would transfer to a different patient population or data source. This absence is a major concern for real-world application, since clinical tools typically require validation on independent cohorts.
6. The paper does not clearly explain how the model was trained and tuned, which affects reproducibility. It's unclear how the train/validation split was handled given the unsupervised nature of the model. It is mentioned that hyperparameters were adjusted, but I don't know how (ie, was there a held-out validation set, or was the entire dataset used for training/clustering?). I'm left unsure whether the reported performance might be inflated by overfitting or trial-and-error hyperparameter tuning on the test set
7. The authors do not provide any error analysis of potential misclustered cases. For a clinical application, understanding whether certain patients were borderline or inconsistently clustered (and why) would be not only important but also interesting. The absence of such analysis means we don't know how robust each assignment is, especially given the small sample size.

**Questions:**

1. What exactly does "multi-" in "multi-model" mean? The term is used repeatedly (and even marked with a special icon in Figure 2), but it isn't explicitly defined.
2. How were hyperparameters tuned and what was the training/validation splits used for model selection? Since this is an unsupervised task, did the authors use a portion of data to validate the representation quality when tuning parameters, or was the entire dataset used for both training and evaluation?
3. What was the rationale for using t-SNE as one of the baseline embedding methods for clustering? t-SNE is stochastic and typically intended for visualization rather than as a fixed embedding for clustering as absolute Euclidean distances could vary across runs, so this was a bit unexpected to me.
4. Given the unsupervised framework and how the graphs are created, it seems to me that this model cannot be directly applied to new patients. So, how were the authors thinking about using this in a clinical setting when applied to new people?
5. Did the authors examine any cases of misclustered or borderline patients, or otherwise measure uncertainty in the cluster assignments? For example, were there patients who repeatedly switched clusters across different runs, or whose embeddings were near cluster boundaries?

---

> ### Author Response · Authors · 2025-11-21
>
> Below are the authors comments on the raised weaknesses and related questions:
>
> 1.      We acknowledge that the dataset of 24 patients is small and limits the generalizability of the findings. This dataset was selected because it was readily accessible for demonstrating the proof of concept. As noted in the final paragraph of the discussion section, the limited sample size is a recognized limitation, and future work will focus on evaluating the approach on larger and more diverse PD patient populations.
>
> 2.      We fully acknowledge that the current model may face scalability challenges when applied to much larger cohorts. As this study was designed as a proof of concept using a small population, scalability was not the primary concern at this stage. In future work, we plan to evaluate the framework on larger datasets and explore architectural optimizations, such as comparing GCNs with more lightweight MLPs and assessing alternative sequential models like LSTMs. to rigorously assess whether the graph-based structure offers meaningful advantages over simpler architectures. These future enhancements are intended to help ensure that the model can be scaled effectively to broader clinical applications.
>
> 3.      We agreed that there was a typo and appreciate the correction. The text has been reviewed to avoid this confusion: The amendments are made in lines [178 – 181]. The term “multi-model” in our original description stemmed from the conceptual idea of using separate components to capture the general structural-temporal embeddings for each node versus local structural embeddings for nodes at each visit-based graph. In the current implementation, these components are integrated into a single model (GCN + GRU generating outputs, then the fusion of GCN and that generated output). We agree that only one model is effectively used in this work. However, our framework is flexible, and in future work it would be possible to explore alternative architectures – for example, using one model for spatio-temporal representations and a different one (such as a spectral model) for structural representations - to further investigate embedding strategies. In response, we have revised the title to more accurately reflect the current scope of the work. We have also removed the ‘multi-model’ reference from the manuscript text and instead included it as a direction for future work. The manuscript is amended to reflect this information, see lines [248-254].
>
> 4.      We acknowledge the fact that t-SNE is stochastic and not a standard embedding method. However, we used t-SNE to assess cluster separability of discovered nonlinear patterns and as an exploratory baseline. However, the primary conclusions rely on the graph-learned embeddings, which are stable, consistent across initialization seeds, and reproducible. See relevant amendments made in lines 344-346
>
> 5.      We totally agree and thank the reviewer for this important point. We acknowledge that our study is limited to a single PD patient cohort and that external validation is necessary for assessing generalizability, and we reflect on this in the last paragraph of the discussion section. However, the current work is intended as a proof of concept to demonstrate the feasibility of learning meaningful patient representations and exploring disease-stage patterns using graph embeddings. Evaluation on external datasets is part of our planned future work, and we agree that such validation is essential before clinical application.
>
> 6.      Our model was trained transductively on the limited size dataset that we have, meaning that all graphs were used to learn the embeddings during training. Hyperparameters, including the embedding dimensions for the GCN, GRU, and output fusion, were tuned via a grid search, using the average adjacency reconstruction loss across visit graphs as the objective. After training, the model was switched to evaluation mode, and the embeddings for each visit graph were extracted. We acknowledge that in a transductive setting no explicit train/validation/test split was used, which is common in unsupervised graph representation learning. This text appears in lines 354 to 358. For future work, we plan to explore inductive settings and external validation to assess generalizability and reduce overfitting. These points are now included in the experiment design section.
>
> 7.      The primary aim of this work is to assess a generic model capable of grouping dynamic patterns in longitudinal records, rather than to provide an immediate clinical application or patient-level interpretation. For this reason, we placed the paper in the “Learning on Graphs and Other Geometries & Topologies” track, rather than a healthcare-specific venue. Your point is valuable and will be considered, especially for larger or external cohorts where error analysis of misclustered cases would be informative. Thank you for the helpful suggestions. See relevant amendments made in lines 486-491.

---

> > ### Comment · Reviewer_QbpR · 2025-11-21
> >
> > I thank the authors for the time taken to answer my review.
> >
> > Unfortunately, I'll have to keep my score. Overall the authors agreed with the limitations/weaknesses I have identified, so even though I appreciate their argument that this is a proof-of-concept (and thus limitations are clearly communicated), I think these weaknesses are too strong for such a specific applied paper, and thus I cannot recommend acceptance. To summarise: (1) I have concerns over the dataset size and scalability issues which impacts the trust for this to be applied on the clinical setting, and (2) from a computational perspective, the idea is interesting but once again I believe the evaluation framework is not strong enough for a conference like ICLR for the reasons I indicated.
> >
> > I hope the authors are able to further strengthen this work as I believe it has a lot of potential and it's just not ready yet to be present at a conference.

---

### Meta-Review · Area_Chair_DNff · 2026-01-09

**Summary:**

The reviewers agreed that the paper is well motivated and practically relevant, with a clear focus on modeling Parkinson’s disease progression using dynamic graph representations, and they appreciated the interpretable GCN–GRU architecture and its reported correlations with clinical scores. However, they consistently identified major shortcomings that outweighed these strengths, including limited methodological novelty, a very small experimental cohort, insufficient and sometimes inappropriate baselines, and multiple issues in experimental rigor and presentation. Concerns about reproducibility, clarity, and the lack of comparisons with recent spatiotemporal graph models further undermined confidence in the work. As a result, despite its application potential, the paper was judged not to meet ICLR’s standards for originality, robustness, and completeness, leading to a rejection decision.

**Reviewer Concerns:**

There are no author rebuttals on the page, making it impossible to assess whether any issues have been addressed. This is a significant omission, as the rebuttal phase should allow authors to clarify misunderstandings or provide additional evidence (such as code link fixes, additional comparisons, or clarification of terminology). Without rebuttals, all reviewer-raised issues remain, including:

Still existing issues: insufficient originality (no new mechanism demonstrated), risk of small-sample generalization, lack of comparisons and ablation, confusing wording, reproducibility concerns (no code, no training details), misuse of t-SNE, and lack of clinical applicability and error analysis. These issues have gone unaddressed, further reinforcing the paper's weaknesses.

If there were rebuttals, I would expect the authors to provide specific supplementary information on these points, but the current situation does not support any positive changes.

**Reviewer Scores:**

Assuming reviewers can participate in further discussion (e.g., during the meta-review phase or additional interaction), I extrapolate potential score changes based on their confidence level and the strength of their opinions. Discussions may include authors providing missing data or clarifications, but given the lack of rebuttal, I assume limited changes and that most reviewers have already taken a firm stance.

Reviewer QbpR (original score: 6): This reviewer has a high confidence level (4) and a relatively balanced opinion. If the authors clarify the definition of "multi-model," provide code, and external validation during the discussion, they might upgrade to 8, as they have already acknowledged the cluster stability. However, if the small dataset issue remains unresolved, the score may remain the same or slightly decrease.

Reviewer 9Jj8 (original score: 0): High confidence level (4), strong rejection, focusing on novelty and missing comparisons. If the discussion introduces recent model comparisons and ablation, they might upgrade to 4. However, if the presentation issues are not fixed, the score is unlikely to exceed 4, as they consider the technical contributions a fundamental flaw.

Reviewer dDiY (original score: 4): Moderate confidence (3). They could raise the score to 6 (close to the threshold) if the discussion strengthens statistical analysis and corrects statements (e.g., PD/Huntington's errors). However, without new evidence on small sample sizes and novelty issues, the score might drop.

Reviewer ZZtF (original score: 2): Highest confidence (5), comprehensive criticism. They could raise the score to 4 if the discussion adds theoretical proofs, complexity analysis, and efficient code. However, given their emphasis on the lack of a novel mechanism, significant changes are unlikely, and the score is likely to remain at 2.

---

### Decision · Program_Chairs · 2026-01-26

Reject